# Alterations in the Gut Fungal Community in a Mouse Model of Huntington's Disease

Geraldine Kong,ᵃ* Kim-Anh Lê Cao,ᵇ Anthony J. Hannanᵃ,ᶜ

ᵃFlorey Institute of Neuroscience and Mental Health, University of Melbourne, Melbourne Brain Centre, Parkville, Australia
ᵇMelbourne Integrative Genomics, School of Mathematics and Statistics, University of Melbourne, Parkville, Australia
ᶜDepartment of Anatomy and Physiology, University of Melbourne, Parkville, Australia

**ABSTRACT** Huntington's disease (HD) is a neurodegenerative disorder caused by a trinucleotide expansion in the HTT gene, which is expressed throughout the brain and body, including the gut epithelium and enteric nervous system. Afflicted individuals suffer from progressive impairments in motor, psychiatric, and cognitive faculties, as well as peripheral deficits, including the alteration of the gut microbiome. However, studies characterizing the gut microbiome in HD have focused entirely on the bacterial component, while the fungal community (mycobiome) has been overlooked. The gut mycobiome has gained recognition for its role in host homeostasis and maintenance of the gut epithelial barrier. We aimed to characterize the gut mycobiome profile in HD using fecal samples collected from the R6/1 transgenic mouse model (and wild-type littermate controls) from 4 to 12 weeks of age, corresponding to presymptomatic through to early disease stages. Shotgun sequencing was performed on fecal DNA samples, followed by metagenomic analyses. The HD gut mycobiome beta diversity was significantly different from that of wild-type littermates at 12 weeks of age, while no genotype differences were observed at the earlier time points. Similarly, greater alpha diversity was observed in the HD mice by 12 weeks of age. Key taxa, including *Malassezia restricta*, *Yarrowia lipolytica*, and *Aspergillus* species, were identified as having a negative association with HD. Furthermore, integration of the bacterial and fungal data sets at 12 weeks of age identified negative correlations between the HD-associated fungal species and *Lactobacillus reuteri*. These findings provide new insights into gut microbiome alterations in HD and may help identify novel therapeutic targets.

**IMPORTANCE** Huntington's disease (HD) is a fatal neurodegenerative disorder affecting both the mind and body. We have recently discovered that gut bacteria are disrupted in HD. The present study provides the first evidence of an altered gut fungal community (mycobiome) in HD. The genomes of many thousands of gut microbes were sequenced and used to assess "metagenomics" in particular the different types of fungal species in the HD versus control gut, in a mouse model. At an early disease stage, before the onset of symptoms, the overall gut mycobiome structure (array of fungi) in HD mice was distinct from that of their wild-type littermates. Alterations of multiple key fungi species were identified as being associated with the onset of disease symptoms, some of which showed strong correlations with the gut bacterial community. This study highlights the potential role of gut fungi in HD and may facilitate the development of novel therapeutic approaches.

**KEYWORDS** DNA sequencing, fungal-bacterial interactions, host-cell interactions, metagenomics, mycology

Address correspondence to Anthony J. Hannan, anthony.hannan@florey.edu.au.

*Present address: Geraldine Kong, Department of Microbiology and Immunology, Peter Doherty Institute for Infection and Immunity, University of Melbourne, Australia.

The authors declare no conflict of interest.

Huntington's disease (HD) is a heritable neurodegenerative disorder that is characterized by gradual brain atrophy, particularly in the cerebral cortex and striatum, although other parts of the brain and body are affected (1). The afflicted individuals suffer from progressive motor, cognitive, and psychiatric symptoms, leading to death

after 15 to 20 years (1). HD is formally diagnosed when motor symptoms become apparent, which typically occurs between 35 and 55 years of age, although other symptoms may manifest 10 to 15 years prior to this (1). The disease is caused by the expansion of CAG repeats in the *huntingtin* (*Htt*) gene, which is ubiquitously expressed not only in the brain but also in many tissues throughout the body, including skeletal muscles and gut intestinal epithelial cells (2–4). The expression of the HD mutation profoundly disrupts transcriptional regulation and normal cellular function, which affects overall physiology (5–7). Thus, the afflicted individuals also suffer from peripheral impairments, including skeletal muscle atrophy, significant weight loss, and impaired immune response (8–11). Recent evidence from our group and others has also shown altered gut bacteria (bacteriome) in HD preclinical models and patients, occurring during the early stage of the disease (12–14).

The gut microbiome encompasses a rich and complex ecosystem composed of various microorganisms, including bacteria, fungi, viruses, and archaea. Studies from the last decade have demonstrated the importance of the gut commensal bacteria in the regulation of host food metabolism and energy homeostasis, as well as the host immunological response. Notably, the gut microbiome maintains a bidirectional communication with the brain through the microbiome-gut-brain axis, and it is capable of regulating host brain health and behavior (15). While previous studies have largely focused on the bacterial residents (i.e., the bacteriome) of the gut, the fungal microbiome (i.e., the mycobiome) has been less explored. The mycobiome represents a relatively small fraction of the gut microbiome (0.01% to 0.1%); however, it has recently begun to gain recognition as an integral part of the gut ecosystem in maintaining host health (16–18).

Similar to intestinal bacteria, commensal fungi are an integral component in host immunomodulation and metabolism (19–22). It has recently been demonstrated that the gut fungal community can induce profound changes in the gut bacteriome structure and shape the gut microbiome during early life (23). These fungal taxa could then elicit robust local and systemic immune responses from synergistic interactions with the gut bacterial population and, thus, play a role in the maturation of the host immune system (23, 24). Additionally, certain fungal species are widely used as probiotics, as they can secrete enzymes that inactivate toxins produced by inflammatory intestinal residents and suppress the growth of other potential pathobionts (25–27). For example, *Saccharomyces boulardii* is prescribed as a probiotic for the management of diarrhea and to prevent intestinal colonization with *Clostridium difficile* following antibiotic therapy (28–30). Oral administration of *Candida kefyr* has been shown to alter the gut microbiome composition and attenuate gastrointestinal inflammation (31). Furthermore, fungi are able to synthesize and release neurotransmitters, similarly to many bacteria. *Saccharomyces cerevisiae* and *Penicillium chrysogenum* can produce high concentrations of norepinephrine, which is capable of reducing anxiety (32, 33). *Candida albicans* has been shown to produce histamine, another neuromodulator involved in appetite regulation, sleep-wake rhythm, and cognitive activity (34). These neurotransmitters could also promote or inhibit bacterial and fungal growth, in addition to their neuromodulatory effects on the enteric nervous system, if not the central nervous system (33).

However, intestinal fungi can also elicit deleterious effects on host health and have been associated with a number of disorders, including gastrointestinal diseases like inflammatory bowel disease (IBD), Crohn's disease (CD), and colorectal cancer (35–38). One study reported increased representation of *Candida albicans* and overgrowth of other Candida species and decreased proportions of *S. cerevisiae* in IBD patients (35). In CD, *Candida tropicalis* was overrepresented in patients and correlated positively with anti-*S. cerevisiae* antibodies (36). Additionally, altered *Basidiomycota/Ascomycota* ratios were observed in these diseases (35, 39). Other recent evidence has shown disruption of the gut mycobiome in neurological disorders, including autism spectrum disorder (ASD), multiple sclerosis (MS), mild cognitive impairment, and Alzheimer's disease (40–45). Moreover, perturbations in the gut fungal-bacterial relationships were observed in these disorders (36, 37, 42, 43, 46).

Remarkably, Alonso et al. reported evidence of yeast-like cells and fungal antigens in postmortem brain slices of HD patients, suggesting that HD patients experience fungal infections more commonly than matched controls (47). Furthermore, the enzyme chitinase 3-like 1 protein, secreted by astrocytes and macrophages, was elevated in the cerebrospinal fluid in HD (48). This enzyme can be activated by chitin, a major component of the fungal cell wall. Given that the gastrointestinal tract is the major source of fungal antigens and infection, we hypothesized that the HD gut mycobiome composition might be altered, which might then contribute to HD pathogenesis and progression.

We have previously shown through shotgun metagenomic sequencing the change in the gut bacteriome profile in the R6/1 transgenic mouse model of HD (herein known as HD mice) at 12 weeks of age, which coincides with the onset of overt motor symptoms (49). Here, we aimed to characterize the gut mycobiome profile in HD mice and determine the unique characteristics that distinguish it from the gut mycobiome profile of healthy controls, as well as to interrogate the fungal-bacterial interactions in these mice.

## RESULTS

The current study investigated the fungal community in the shotgun metagenomic data set constructed from 90 fecal samples of R6/1 HD transgenic mice and their wild-type (WT) littermates sampled at five time points: 4 (young), 6, 8, 10, and 12 (early disease stage) weeks of age (49). On average, 0.1% of the reads aligned to fungal genomes, which primarily consisted of fungi from the *Ascomycota* (~61%) and *Basidiomycota* (~22%) phyla, with *Ascomycota* being the most abundant phylum represented. The most abundant genera in the murine gut fungal genera consisted of *Aspergillus* (~6%), *Batrachochytrium* (~3.27%), *Schizosaccharomyces* (~2.15%), and *Fusarium* (1.7%).

**HD mice harbor an altered gut mycobiome composition at week 12.** The *Ascomycota/Basidiomycota* ratio was previously reported as an indicator of healthy gut fungal composition. In our case, no significant differences in these two phyla were observed between the two genotypes. The Shannon alpha diversity index, which measures the evenness and richness of communities within a sample, did not differ between the two genotypes during early life until the young adult age, from week 4 to week 10, but it increased in the HD gut at week 12 (*t* test, $P = 0.034$) (Fig. 1A). To determine the variability in the gut fungal communities, the average distance of each sample to the group centroid was calculated and compared. Overall, the fecal mycobiome was less heterogenous in HD mice than in their WT littermates at week 6 ($P = 0.018$) and week 8 ($P = 0.005$) but not at any other time points (Fig. 1B). Additionally, mycobiome stability was determined by calculating the Aitchison distance travelled by each sample across time, as previously described (50). There were no significant differences in within-subject fungal stability between WT and HD mice (see the supplemental material). The beta diversity of gut mycobiome was similar between the two genotypes from week 4 to week 10. In contrast, the gut mycobiome composition of HD mice was significantly different from that of their WT counterparts at week 12, as shown by the clustering of samples according to genotype in principal-component analysis (PCA) (permutational multivariate analysis of variance [PERMANOVA], $P = 0.025$) (Fig. 1C).

As an overall difference in gut fungal communities was observed only at week 12, the supervised sparse partial least-squares discriminant analysis (sPLS-DA) method was employed to identify the main species driving this difference. A signature consisting of 15 fungal species that were discriminatory between the two genotypes was identified (classification error rate = 0.27) (Fig. 2). In particular, the fungal species identified as having the best predictive value were *Glarea lozoyensis*, *Malassezia restricta*, *Penicillium digitatum*, *Yarrowia lipolytica*, and *Aspergillus* spp., including *Aspergillus fischeri*, *Aspergillus uvarum*, and *Aspergillus alliaceus*, all of which were depleted in the HD mouse mycobiome compared to their levels in the WT littermate mycobiome. Furthermore, *Penicillium solitum* and *Meyerozyma guilliermondii* were also selected in the signature, both of which were enriched in the HD gut mycobiota.

To understand the potential changes in relative abundances underlying the temporal shift in the gut mycobiome, linear mixed-effect model splines (LMMS) modeling

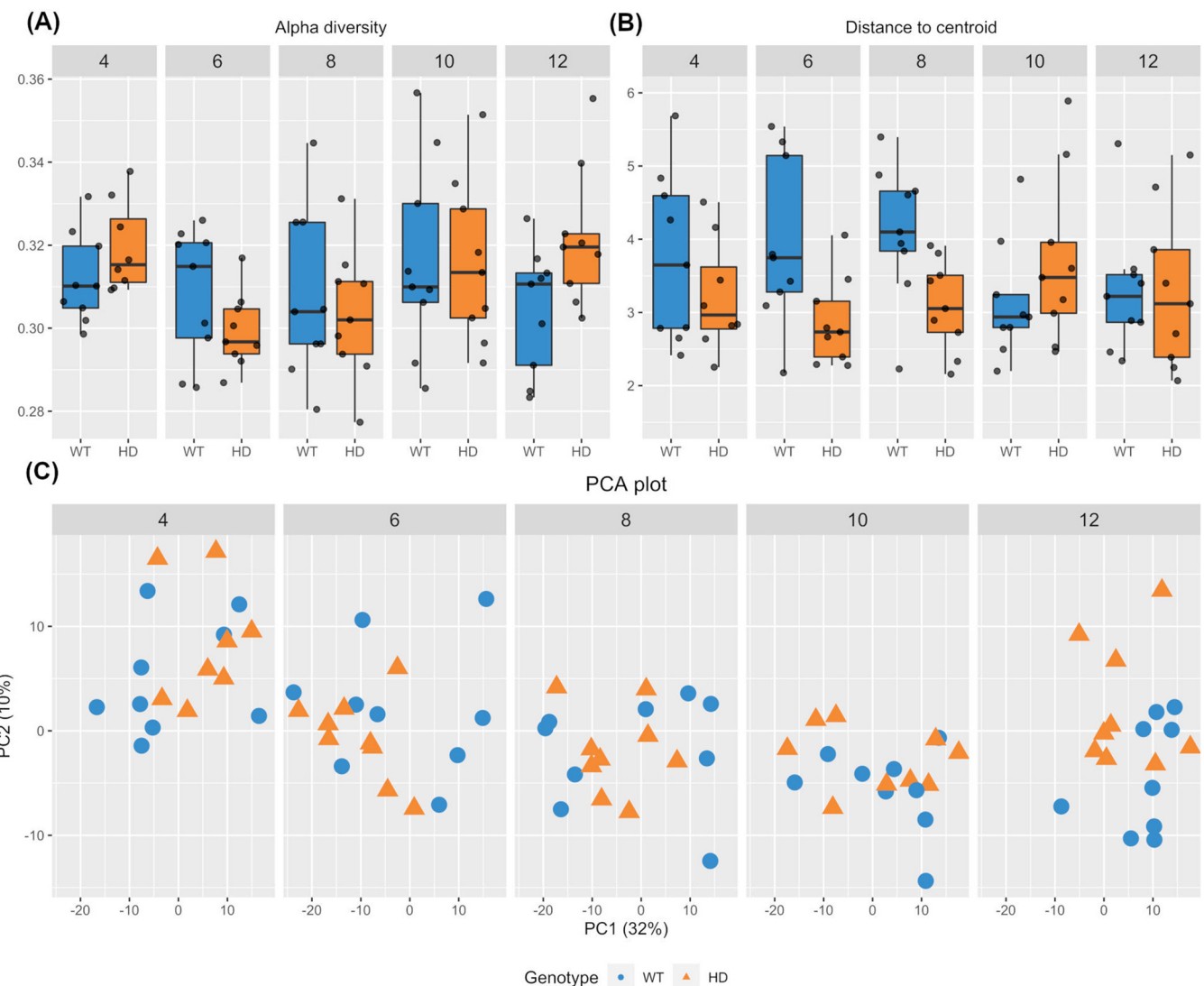

**FIG 1** Gut mycobiome alterations in HD mice. (A) Increased alpha diversity of fecal mycobiome in HD mice at week 12, but not any earlier age, compared to their WT littermates (Kruskal-Wallis, $P = 0.034$). (B) Boxplot showing the distance of each sample to the group centroid. HD mice displayed less heterogeneity in gut mycobiome composition only at week 6 ($P = 0.018$) and week 8 ($P = 0.005$). (C) Principal-component analysis (PCA) plot of the gut mycobiome. The HD mouse gut mycobiome composition was significantly different from that of their WT littermates at week 12 but not at any earlier time point (PERMANOVA, $P = 0.025$).

and sparse principal-component analysis (sPCA) implemented in the TimeOmics package were applied. *Candida glabrata*, *Candida parapsilosis*, and *Cladophialophora carrionii* displayed coabundance and correlated negatively with *P. digitatum* and *Paracoccidioides lutzii*, which was confirmed by the proportionality test (see the supplemental material). Notably, the temporal trends of *P. solitum*, *M. guillerimondii*, *C. glabrata*, and *C. parapsilosis* were different between the two genotypes (Fig. 3). *P. solitum* and *C. glabrata* gradually decreased over time in the WT mice, but their relative abundances remained unchanged in HD mice (Fig. 3). On the other hand, *M. guilliermondii* and *C. parapsilosis* remained at stable levels in the WT mice, while they increased gradually in the HD mice (Fig. 3). However, differential-abundance analysis of all LMMS profiles revealed no significant effect of genotype or interaction effect of genotype and time in any of these longitudinal profiles (see the supplemental material).

**Integration of bacterial and fungal data sets hints at the perturbation of interkingdom interaction in HD mice.** Given that the majority of the changes in gut fungal community structures were observed at the week 12 time point, we focused on this time point to assess potential associations between the gut fungal and bacterial

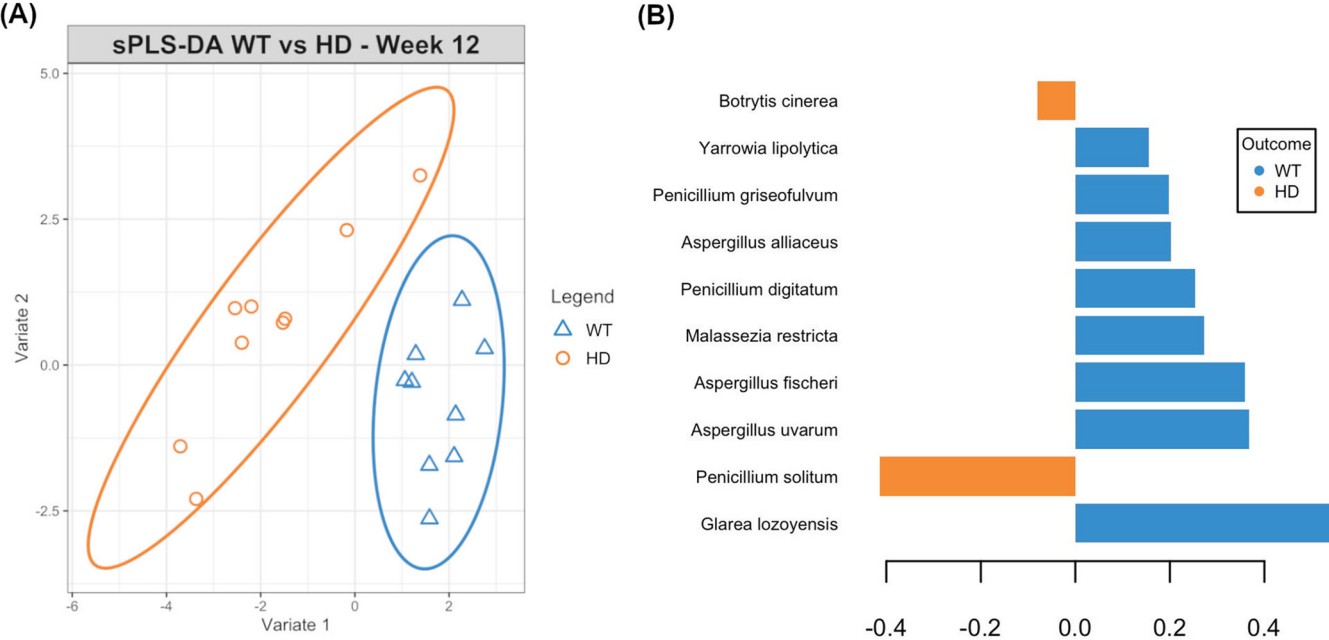

**FIG 2** Signature gut fungal species discriminatory between WT and HD mice identified by sparse PLS-DA (classification error rate = 0.27). (A) Sample plot (95% confidence ellipses) revealed distinct clustering of WT and HD samples, indicating the predictive power of the selected fungal signature to distinguish between the two genotypes. (B) Contributions of the top 10 fungal species identified to be discriminatory between WT and HD samples. Species with negative contributions were enriched in HD mice (largest mean value in this group), while those with positive contributions were enriched in WT mice.

communities. The gut mycobiome data set here was integrated with the previously published gut bacterial data set using DIABLO from the mixOmics R package. With a design matrix of 0.1, which was set to maximize the discrimination between two genotypes, the correlation between the two data sets was still very strong ($r = 0.89$) (Fig. 4A). Feature selection identified that the fungal species *Glarea lozoyensis*, *M. restricta*, *A. alliaceus*, and *Y. lipolytica* all possessed strong positive associations ($r > 0.6$) with the bacterial *Bacteroides* spp., including *Bacteroides pyogenes*, *Bacteroides oleiciplenus*, and *Bacteroides timonensis*, and with *Prevotella scopos* (Fig. 4C). Additionally, *Lactobacillus reuteri* was negatively associated with *G. lozoyensis*, *M. restricta*, *A. alliaceus*, and *Y. lipolytica* ($r < -0.5$). *A. fischeri* was found to positively associate with only *Prevotella scopos* ($r = 0.65$) and *Bacteroides pyogenes* ($r = 0.63$). Notably, all the selected fungal species were depleted in the HD gut mycobiome.

## DISCUSSION

This study presents the first evidence of the disruption of the gut fungal composition in Huntington's disease. At 12 weeks of age, which corresponds to an early disease stage and is prior to overt motor symptoms, there was a significant difference in beta diversity between HD mice and their WT littermates that was not observed at any earlier age (51). The HD mice also showed a less heterogenous gut mycobiome configuration than healthy controls at weeks 6 and 8. However, there were more volatile fungal taxa in HD mice. Reduced heterogeneity of the gut microbiome has been observed in a number of diseases, and our present data suggest the presence of particular stressors in HD mice that induce similar ecological changes in gut fungal communities across individuals at these time points (52, 53). An increase in alpha diversity was also observed in HD mice at 12 weeks of age without any alterations at earlier ages. Changes in gut mycobiome alpha diversity can be challenging to interpret, as both elevated and reduced alpha diversity have been associated with adverse health outcomes (43, 45, 46). Therefore, rather than lacking fungal diversity, it is possible that the affected individuals may harbor fewer commensals in favor of proinflammatory pathobionts.

Different longitudinal profiles between the two genotypes were observed for gut commensals *Candida parapsilosis*, *Candida glabrata*, *Meyerozyma guilliermondii*, and

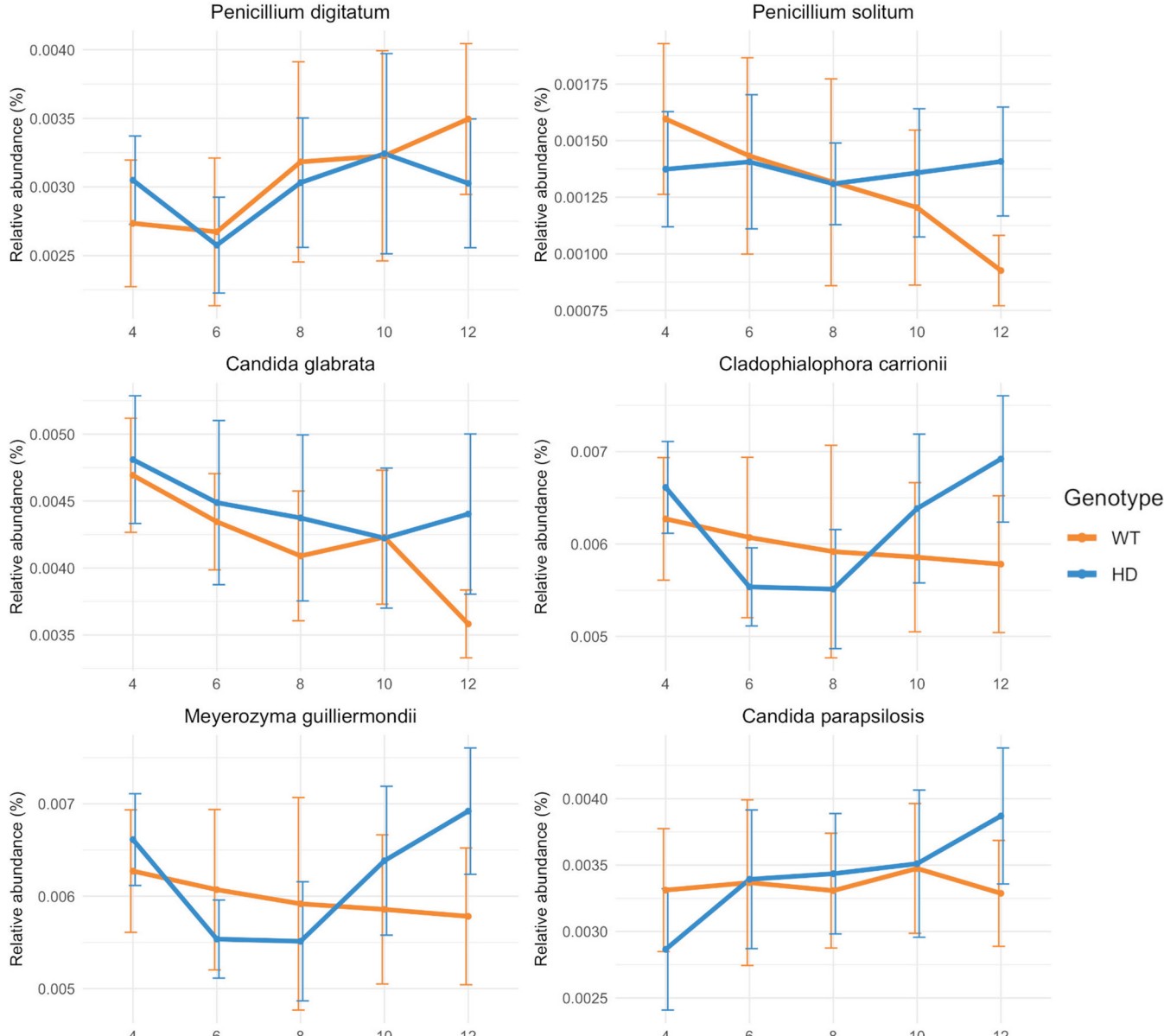

**FIG 3** Temporal relative abundance of coabundant fungal taxa with different trends between samples from WT and HD mice across time. Mean values and standard deviations are shown. $n$ = 9/group.

*Penicillium solitum*, with increasing prevalence in the HD gut over time. *C. parapsilosis* and *C. glabrata* are both opportunistic fungal pathobionts commonly found in the gut that can have devastating effects, particularly in immunocompromised patients, where they can cause candidiasis and increased mortality (54, 55). *C. glabrata* and *C. parapsilosis* overgrowth showed no adverse outcomes in healthy mice but exacerbated gastro-inflammation in mice with dextran sulfate sodium (DSS)-induced colitis via stimulation of proinflammatory cytokines, receptor expression, and shifting of major immune cell types (23, 56, 57). Concurrent with our results, enrichment of *C. glabrata* and *C. parapsilosis* was reported in the gut mycobiomes of patients with ASD, IBD, and Crohn's disease (46, 58–60). Individuals with elevated antigens of *C. parapsilosis* and *C. glabrata* were associated with higher risk of MS (61). Moreover, it was recently reported that HD patients had fungal structures with *C. glabrata* antigens in corpora amylacea (CA) in the brain, suggesting that the infection by this fungus occurred at an earlier period long before death, as its secreted proteins were trapped during the formation of CA

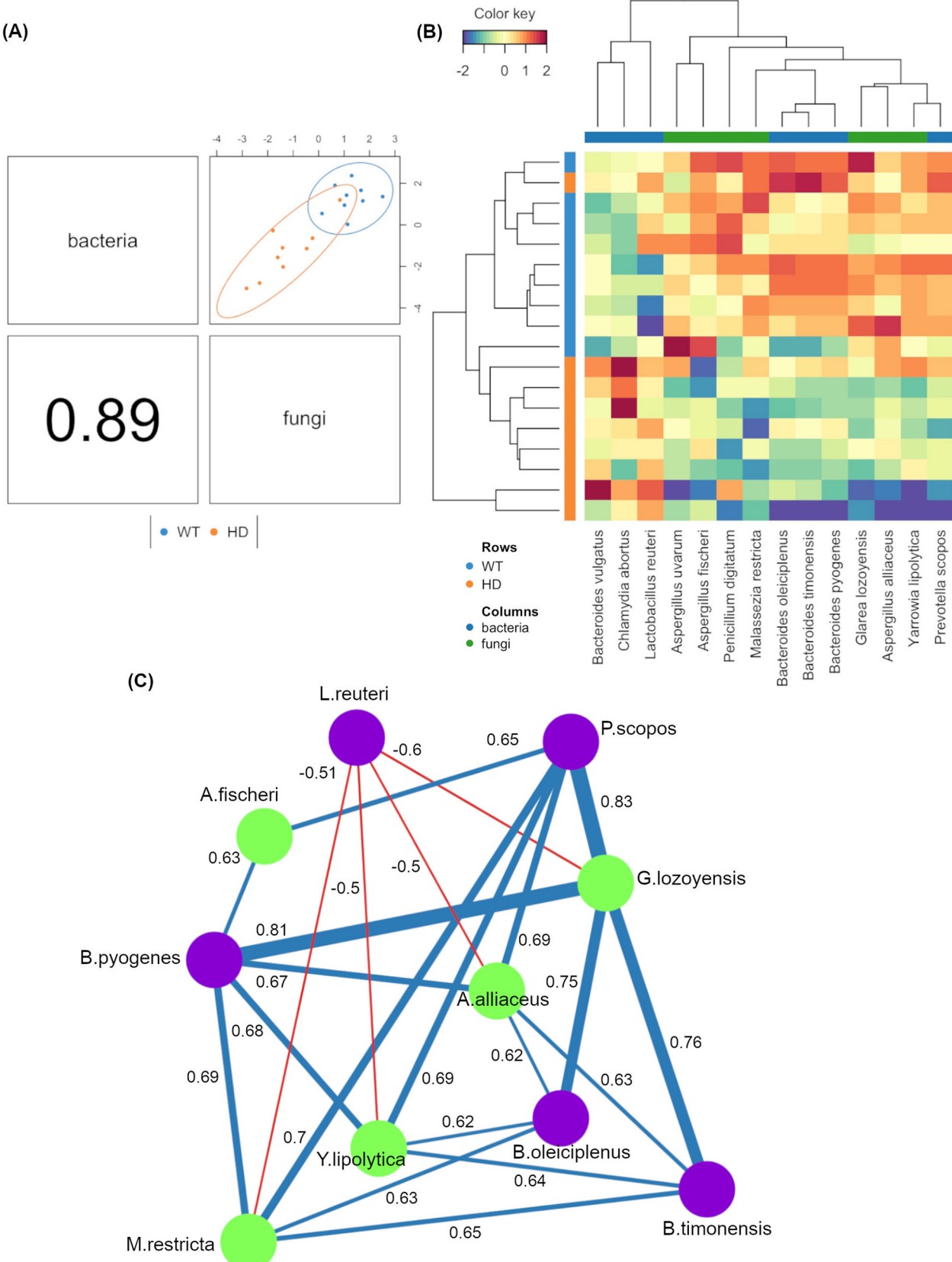

**FIG 4** Integration of gut bacteriome and mycobiome compositions using DIABLO. (A) The gut bacteriomes and mycobiomes were highly concordant with each other (*r* = 0.89) and revealed distinct clustering of samples from WT and HD mice. (B) Clustered heatmap showing the

over the years (47). Increased gut inflammation and permeability, as well as impaired blood-brain barrier, are also comorbidities of HD (62–66). Taken together, our results suggest a bloom in *C. glabrata* that may aggravate gastrointestinal inflammation in HD and lead to candidemia and *Candida* infection in the HD brain.

Furthermore, alterations in the abundance of several key fungal species were identified. Enrichment of *Penicillium digitatum* and *Botrytis cinerea* was observed in the HD gut, in line with a previous study reporting the presence of *Botrytis cinerea*, which may have originated from the intestine, in several brain regions in patients with Alzheimer's disease (67). On the other hand, *Glarea lozoyensis* was more prevalent in healthy controls. *G. lozoyensis* is a natural producer of the antimicrobial compound pneumocandin B0, which is capable of inhibiting the growth of fungal pathobionts like *C. albicans* (68, 69). Furthermore, *Malassezia restricta* and *Yarrowia lipolytica* were underrepresented in the HD gut mycobiome at 12 weeks of age. Although *Y. lipolytica* is commonly used in the food industry to produce long-chain polyunsaturated fatty acids, it has also been observed in mammalian stool samples and its potential as a probiotic was recently demonstrated (70, 71). Oral supplementation of *Y. lipolytica* considerably reduced the bacterial load and improved the host immunological response against bacterial infection in fish (72). In turkeys and pigs, dietary administration of this fungus led to enhanced cytokine production and blood leukocyte counts, increased mucosal immunity, and improved overall gut architecture (73–75). As such, supplementation of *Y. lipolytica* may be beneficial in ameliorating some of the HD symptoms.

*Malassezia* is a dominant commensal found on the skin but also one of the most prevalent fungal genera in the human gastrointestinal tract, associated with intestinal mucosa (18, 60, 76–79). *M. restricta*, in particular, is associated with Crohn's disease and ulcerative colitis (80). Colonization of *M. restricta* in germ-free mice exacerbated the severity of dextran sodium sulfate-induced colitis (80). The presence of *M. restricta* stimulated Card9 signaling via the Dectin-2 receptor, which then induced proinflammatory responses toward innate immune cell activation and aggravated intestinal inflammation (80). Recent work demonstrated that fungal signals alone are unable to induce the level of colonic inflammation commonly reported in DSS-treated animals (23), and hence, this activation is potentially dependent on the concomitant presence of bacterial signals.

Our bacteriome-mycobiome integration analysis revealed the negative correlation of *Lactobacillus reuteri* with a number of fungal species, including *M. restricta*, *Y. lipolytica*, *G. lozoyensis*, and *Aspergillus* spp., which were identified as differentially abundant between the two genotypes. In line with this, several studies have shown the opposing relationship between *L. reuteri* and fungal colonization (23, 81–83). Reuterin, a main metabolite secreted by *L. reuteri*, is a well-known antimicrobial substance that can also suppress fungal growth (83, 84). Additionally, lactic acid and organic acids are also products of *L. reuteri* that could decrease gut pH and impact fungal growth (85). Apart from that, *L. reuteri* has neuromodulatory effects in the enteric nervous system since it is capable of producing the neurotransmitter gamma-aminobutyric acid (GABA), which could also directly impact the growth of other residents in the gut (33, 86). Given that a delicate bacteriome-mycobiome balance is required to maintain host health and homeostasis, the altered interkingdom equilibrium in HD could contribute to or modulate gastrointestinal inflammation or the microbiome-gut-brain axis in HD.

The current study investigating the HD gut mycobiome profile is the first to be conducted; however, it possesses several caveats. This was a pilot study with a small sample size, constrained by the cost of metagenomics sequencing, and hence, generalizable extrapolation from it may be limited. Furthermore, the sequencing depth of the investigated shotgun metagenomics data may be insufficient, considering the extremely low abundance of

**FIG 4** Legend (Continued)
abundances of bacterial and fungal species that were found to covary with each other (columns) and were discriminatory between the two genotypes (rows). (C) Correlation network analysis revealed strong positive- and negative-correlation associations between several bacterial and fungal taxa. Green and purple nodes correspond to fungi and bacteria, respectively. Correlations of $r > 0.5$ (blue lines) or $r < -0.5$ (red lines) are shown.

fungi in stool, which may affect the accuracy of the results. The ability of fungi to change morphologically depending on environmental conditions can also often lead to misidentification in fungal databases. In addition, many of the fungal species detected are of environmental origin, and their biological relevance in the mammalian gut is unknown. Nevertheless, the results presented here generated novel hypotheses that could inform future studies aiming to understand the microbiome-gut-brain axis in HD.

In summary, this is the first evidence of the perturbation of gut mycobiome composition in HD. Alterations in the fungal composition and structure of the gut mycobiome were observed at the onset of motor symptoms in HD mice. These changes were characterized by depletion of several key species in the HD gut, including *G. lozoyensis*, *Aspergillus uvarum*, *Aspergillus fischeri*, *M. restricta*, and *P. digitatum*. Moreover, the abundance of *Lactobacillus reuteri* was negatively correlated with those key fungal taxa, suggesting perturbation of interkingdom interactions within the HD gut. The study presented here highlights the need to examine the HD gut mycobiome in more depth, which would advance the understanding of gastrointestinal fungal-bacterial relationships important for host health. This may allow us to identify novel therapeutics for HD and related disorders.

## MATERIALS AND METHODS

**Taxonomic classification of fungal genomic reads.** Shotgun metagenomic sequences of male R6/1 HD transgenic mice and their wild-type (WT) littermates were previously deposited by our group in NCBI (BioProject accession number PRJNA613182). The data were from 90 stools from 18 mice sampled longitudinally at 5 time points: weeks 4, 6, 8, 10, and 12. Reads that aligned to the mouse genome (mm10) were removed to obtain host-contaminant-free reads with approximately 10,389,634 read pairs of 251 bp per sample. To identify the taxonomic composition, reads were classified using Kaiju 'Greedy' mode and matching to the NCBI BLAST nonredundant plus eukaryotic protein database (87). Read counts assigned to the fungal kingdom were then extracted for analysis.

**Diversity analysis of fungal DNA reads.** For alpha diversity analysis, data were rarefied to the 4,863,549 sequences per sample before calculating the Shannon index using the phyloseq R package (88). The data were stratified to their age groups before testing for the significance of genotype with the *t* test, after confirming the equality of variance and normality of the data through Levene's test and the Shapiro-Wilk test, respectively (89, 90). The *P* values were then adjusted for multiple comparisons with false discovery rate (FDR) (91). Beta diversity analysis was performed by calculating the Aitchison distance and visualizing it with principal-component analysis (PCA). Similarly, the data were stratified to their respective age groups and then tested for the effect of genotype using Adonis PERMANOVA from the vegan R package (92). The *P* values were then corrected for multiple testing using FDR. To ascertain the variability in the overall gut mycobiome structure, the distance of each sample from the group centroid was calculated and its significance was determined using the functions *betadisper*() and *permutest*(), respectively (92). In all cases, the significance level was set at <0.05.

To identify the fungal signature discriminating between the two genotypes, sparse partial least-squares discriminant analysis (sPLS-DA) from the mixOmics R package was performed on the centered log ratio (CLR)-transformed data (93, 94). The parameters chosen to select a signature of 15 fungal species and "leave-one-out" cross-validation were used to evaluate the classification performance of the method.

**Clustering analysis and volatility assessment of fungal temporal profiles.** Analysis of the longitudinal fungal community structure was performed using the TimeOmics R package (95, 96). Briefly, each feature was fit with the best possible model using linear mixed-effect model splines (LMMS) to cluster the fungal species across time (97). This method accounts for between- and within-individual variability and irregular time sampling while avoiding under- or oversmoothing. The Breusch-Pagan test and mean square error were used to assess the homoskedasticity of residuals to remove profiles that were considered noisy before clustering. Sparse PCA (sPCA) was applied on the smooth profiles to identify and cluster species that contributed to explaining time variability. To determine the significance of differences in each temporal profile between the two genotypes, LMMS for differential expression analysis (LMMSDE) was performed (97). As correlations can be spurious for microbiome data that are compositional, the proportionality distance between species identified as relevant in our method was calculated as a measure of association. Median distances within clusters were calculated and compared to all the median distances between all species.

**Supervised integration for interkingdom interaction.** Bacterial counts were extracted from the Kaiju taxonomic assignment output. To identify the associations between the gut bacteriome and mycobiome that were altered in HD, CLR-transformed fungal and bacterial data sets were integrated using the DIABLO framework from the mixOmics package in a supervised analysis (94, 98). A design matrix of 0.1 was used to place higher importance on the discrimination of genotypes rather than maximizing correlation between the two data sets. A signature of 7 bacterial and 7 fungal species was selected for the model. A correlation cutoff of 0.5 was set for the network visualization using Gephi (99).

## SUPPLEMENTAL MATERIAL

Supplemental material is available online only.

**SUPPLEMENTAL FILE 1**, XLSX file, 0.05 MB.

## ACKNOWLEDGMENTS

We thank Thibault Renoir for his advice and assistance regarding the HD mouse model.

This research received no specific grant from any funding agency in the public, commercial, or not-for-profit sectors. We acknowledge that A.J.H. was supported in part by a National Health and Medical Research Council (NHMRC) Principal Research Fellowship (grant number GNT1117148) and the DHB Foundation (Equity Trustees), and K.-A.L.C. was supported in part by an NHMRC career development fellowship (grant number GNT1159458). The shotgun sequencing was funded by a Computational Biology Research Initiative (CBRI) grant to K.-A.L.C. and A.J.H. from the University of Melbourne.

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
