## [Reviewer comments · Microbiology Spectrum]

Microbiology Spectrum

Alterations in the gut fungal community in a mouse model of Huntington's disease

Anthony Hannan, Kim-Anh Lê Cao, and Geraldine Kong

Corresponding Author(s): Anthony Hannan, Florey Institute, University of Melbourne

Review Timeline:

Submission Date:	November 9, 2021
Editorial Decision:	January 17, 2022
Revision Received:	February 9, 2022
Accepted:	February 14, 2022

Editor: Rebecca Shapiro

Reviewer(s): The reviewers have opted to remain anonymous.

Transaction Report:

DOI: <https://doi.org/10.1128/spectrum.02192-21>

January 17, 2022

Prof. Anthony John Hannan
Florey Institute, University of Melbourne
Melbourne
Australia

Re: Spectrum02192-21 (Gut fungal dysbiosis in Huntington's disease)

Dear Prof. Anthony John Hannan:

Thank you for submitting your manuscript to Microbiology Spectrum. Two reviewers have read and reviewed your manuscript, and we find that only minor modifications will be needed for possible publication of your manuscript. In particular, please pay attention to comments pertaining to statistical analysis. We look forward to seeing a revised copy soon!

Link Not Available

Sincerely,

Rebecca Shapiro

Journals Department
Reviewer comments:

Reviewer #1 (Comments for the Author):

In this study, the authors provides evidence of an altered gut fungal community (mycobiome) in the R6/1 transgenic mouse model of HD. This a very-well written study that provides novel data to the field.

Although the authors used a relatively small number of animals (n=9 /genotype), the longitudinal collection of fecal samples from 4- to 12-weeks is very interesting, as it offers the opportunity to correlate individual mouse "disease trajectory" with changes in the mycobiome. However, no attempt was made to correlate any behavioral abnormalities with changes in the mycobiome (e.g., alpha diversity)?

The authors should modify the title and clarify "in a mouse model of Huntington's disease".

The authors should include other information about the animals (e.g., sex, presumably males?, weight at each of the 5 points).

What is driving the changes at 12 weeks of age? Previous studies have shown gut inflammation in the same mouse model

during adulthood. What is the relationship between age, gut inflammation, motor dysfunction and alterations in the mycobiome?

Reviewer #2 (Public repository details (Required)):

The Microbiome raw sequencing data should be submitted and the link should be clearly findable in the paper.

Reviewer #2 (Comments for the Author):

This is a manuscript on the role of the bacteriome and mycobiome in a mouse model of huntington's disease. The authors find some alterations in the micorbiome and mycobiome in the mice with HD, which is unsurprising given the genetic nature of the disease. Generally, the text is well-written. The authors use long bold headers for their paragraphs, which I found helpful to quickly pick up the message. The bioinformatics analysis is reasonable and the conclusions follow from the results. Generally, the manuscript is in good shape. I have written down some comments below.

Major comments:

Huntington's disease is a relatively well-understood disease with a clear genetic origin. Because of this, it would be very interesting to take this analysis to a functional level and see if any genes in the microbiome that are in any way related to Huntington's disease are altered in the bacteriome and/or mycobiome.

It is not clear why the authors used a Kruskal Wallis test to determine an increase in HD mice at week 12. I see the argument for using a non-parametric test as variance is unequal over time, but KW is typically used for more than two groups. If in fact the KW was ran over all 10 groups, a post-hoc is necessary to determine differences between two groups.

Similarly, for the PERMANOVA in fig1C, is this a test for the entirety of the data or just for the HD vs WT of that age group?

It is likely a post-hoc correction as well as post-hoc test would be warranted in both cases.

Minor comments:

The authors analyse longitudinal micorbiome data and briefly investigate the temporal stability of several taxa. Change over time in the microbiome is known as volatility and in their previous work (10.1016/j.nbd.2020.105199), the authors do indeed measure bacteriome volatility in the same animals (though they did not refer to the original manuscript describing volatility in either of their manuscripts: 10.1016/j.psyneuen.2020.105047). Why do the authors not perform the same analysis here?

The authors use the term dysbiosis. This term has lost favour in the field as it is hard to define. (A microbiome does not have to be unhealthy for it to make the host sick) Please rephrase.

Typos:

Line 146 "HD mice harbours" should be harbour.

Staff Comments:

Preparing Revision Guidelines

Please return the manuscript within 60 days; if you cannot complete the modification within this time period, please contact me. If you do not wish to modify the manuscript and prefer to submit it to another journal, please notify me of your decision immediately so that the manuscript may be formally withdrawn from consideration by Microbiology Spectrum.

Response to Reviewers

We thank the Editor for obtaining constructive peer review and the Reviewers for taking the time to share their expertise and help us improve our manuscript. We have extensively revised our manuscript to address each and every comment from the Reviewers. We detail our responses below, and in the revised manuscript. Many thanks for your kind assistance and consideration.

Reviewer #1

In this study, the authors provides evidence of an altered gut fungal community (mycobiome) in the R6/1 transgenic mouse model of HD. This a very-well written study that provides novel data to the field.

Although the authors used a relatively small number of animals (n=9 /genotype), the longitudinal collection of fecal samples from 4- to 12-weeks is very interesting, as it offers the opportunity to correlate individual mouse "disease trajectory" with changes in the mycobiome. However, no attempt was made to correlate any behavioral abnormalities with changes in the mycobiome (e.g., alpha diversity)?

RESPONSE: As behavioural tests have been shown to affect gut microbiome composition, we kept this cohort of mice as behaviourally naïve as possible aside from rotarod testing, which assesses motor coordination. We did not correlate this with changes in mycobiome alpha diversity, as alpha diversity is challenging to interpret since both increased and decreased alpha diversity has been associated with negative health outcomes. However, we did integrate the longitudinal mycobiome dataset with longitudinal rotarod data. Since many of mice did not show any significant impairments in rotarod performance by Week 12, the integration resulted in minimal correlations and very poor performance of the machine learning model. Similar results were obtained when integrating body weight data with mycobiome composition. Thus, we did not include these results.

The authors should modify the title and clarify "in a mouse model of Huntington's disease".

RESPONSE: The title has been amended.

The authors should include other information about the animals (e.g., sex, presumably males?, weight at each of the 5 points).

RESPONSE: Yes, male mice were used in the study. The weight of each mouse at the 5 time points and motor performance based on rotarod have been included in the Supplementary Data.

What is driving the changes at 12 weeks of age? Previous studies have shown gut inflammation in the same mouse model during adulthood. What is the relationship

between age, gut inflammation, motor dysfunction and alterations in the mycobiome?

RESPONSE: We agree it is a very important question. Huntington's disease is caused by a genetic mutation in the *huntingtin (HTT)* gene which is also expressed in the gut. We believe that the changes in the gut microbiome is primarily due to expression of the genetic mutation, which could alter gut structure, function and induce gut inflammation, resulting in gut microbiome alteration. Given that Huntington's disease is a progressive disease, it is unknown exactly at what age does gut inflammation and significant damage to the gut architectural structure occur. However, this is beyond the scope of this study as the aim of it is to examine the gut mycobiome composition and its relation to the gut bacteriome.

Reviewer #2

The Microbiome raw sequencing data should be submitted and the link should be clearly findable in the paper.

RESPONSE: As this is an analysis of an already published dataset, the raw sequencing data is available on the NCBI portal and can be searched using the BioProject Number PRJNA613182 provided in the Methods section (Line 320).

This is a manuscript on the role of the bacteriome and mycobiome in a mouse model of huntington's disease. The authors find some alterations in the micorbiome and mycobiome in the mice with HD, which is unsurprising given the genetic nature of the disease. Generally, the text is well-written. The authors use long bold headers for their paragraphs, which I found helpful to quickly pick up the message. The bioinformatics analysis is reasonable and the conclusions follow from the results. Generally, the manuscript is in good shape. I have written down some comments below.

Major comments:

Huntington's disease is a relatively well-understood disease with a clear genetic origin. Because of this, it would be very interesting to take this analysis to a functional level and see if any genes in the microbiome that are in any way related to Huntington's disease are altered in the bacteriome and/or mycobiome.

RESPONSE: The functional analysis of this shotgun sequencing data and its relation to the changes in HD gut bacteriome has been investigated and published previously (10.1016/j.nbd.2020.105199). We did not apply the same analysis here because the fungal database (mycobiome) is more sparsely annotated compared to its bacteriome counterpart and there are many unculturable fungi yet to be discovered with various unknown function and taxonomy. Additionally, fungi are often mislabelled in databases which complicates taxonomy assignment. Thus, additional functional relationships to the mycobiome will not be reliable, and could be misleading.

It is not clear why the authors used a Kruskal Wallis test to determine an increase in HD mice at week 12. I see the argument for using a non-parametric test as variance is unequal over time, but KW is typically used for more than two groups. If in fact the KW was ran over all 10 groups, a post-hoc is necessary to determine differences between two groups.

RESPONSE: We have amended our analyses using t-tests. For clarification, t-tests were performed to compare WT and HD at each time point and the reported p-values have been adjusted for multiple hypothesis testing using FDR.

This has been clarified at Line 335-338:

“The data was stratified to their age groups before testing for the significance of Genotype with t-tests, after confirming the equality of variance and normality of the data through Levene’s Test and Shapiro-Wilk test respectively (89, 90). The p-values were then adjusted for multiple comparisons with False Discovery Rate (FDR) (91).”.

Similarly, for the PERMANOVA in fig1C, is this a test for the entirety of the data or just for the HD vs WT of that age group? It is likely a post-hoc correction as well as post-hoc test would be warranted in both cases.

RESPONSE: Similar to the alpha diversity test, we first subset the data to their respective age group before testing for Genotype effect with PERMANOVA (there are only two genotypes to test in this study). The resulting p-values were then corrected for multiple hypothesis testing using FDR. As such, the p-value shown in Fig1C has been FDR-adjusted.

We have clarified this at Line (338-343):

“Beta diversity analysis was performed by calculating Aitchison distance with Principal Component Analysis (PCA) visualisation. Similarly, the data was stratified to their respective age groups and then tested for the effect of Genotype using Adonis PERMANOVA from the vegan R package (91). The p-values were then corrected for multiple testing using FDR.”.

Minor comments:

The authors analyse longitudinal micorbiome data and briefly investigate the temporal stability of several taxa. Change over time in the microbiome is known as volatility and in their previous work (10.1016/j.nbd.2020.105199), the authors do indeed measure bacteriome volatility in the same animals (though they did not refer to the original manuscript describing volatility in either of their manuscripts: 10.1016/j.psyneuen.2020.105047). Why do the authors not perform the same analysis here?

RESPONSE: It is known that the gut mycobiome is known to be more unstable than its bacteriome counterparts with many fungal species passing through the GI tract without becoming stable colonizers (<https://doi.org/10.3389/fmicb.2019.01575>), and thus not detected in the same individual over the course of time. Hence, we did not perform the same volatility analysis as in our previous work because those species would be considered as noise by the method and filtered out. Coupled with the low abundance of fungal species in the metagenomic dataset, we deemed that such analysis will not yield reliable results. We did perform volatility analysis as per **10.1016/j.psyneuen.2020.1050** whereby total Aitchison distance travelled by each sample across time was calculated and tested for significance with t-test. In summary, no difference between the two groups were found.

The results have been included in the manuscript at Line 157-160:

“Additionally, mycobiome stability was determined by calculating the Aitchison distance travelled by each sample across time as previously described (50). We found no significant differences in the stability of fungal communities between WT and HD (Supplementary Data).”

The authors use the term dysbiosis. This term has lost favour in the field as it is hard to define. (A microbiome does not have to be unhealthy for it to make the host sick) Please rephrase.

RESPONSE: We have corrected this term throughout the manuscript: in the Title, Line 42 and Line 149.

Typos:

Line 146 "HD mice harbours" should be harbour.

RESPONSE: We thank the Reviewer for flagging this. This has been corrected.

February 14, 2022

Prof. Anthony John Hannan
Florey Institute, University of Melbourne
Melbourne
Australia

Re: Spectrum02192-21R1 (Alterations in the gut fungal community in a mouse model of Huntington's disease)

Dear Prof. Anthony John Hannan:

Your manuscript has been accepted, and I am forwarding it to the ASM Journals Department for publication. You will be notified when your proofs are ready to be viewed.

Sincerely,

Rebecca Shapiro
Editor, Microbiology Spectrum

Journals Department
Supplemental Dataset: Accept